# Targeting FGFRs Using PD173074 as a Novel Therapeutic Strategy in Cholangiocarcinoma

**DOI:** 10.3390/cancers15092528

**Published:** 2023-04-28

**Authors:** Brinda Balasubramanian, Kiren Yacqub-Usman, Simran Venkatraman, Kyaw Zwar Myint, Jitlada Juengsamarn, Poowanai Sarkhampee, Nithi Lertsawatvicha, Jittiyawadee Sripa, Thiti Kuakpaetoon, Chinnawut Suriyonplengsaeng, Kanokpan Wongprasert, Anna M. Grabowska, David O. Bates, Tavan Janvilisri, Rutaiwan Tohtong

**Affiliations:** 1Department of Biochemistry, Faculty of Science, Mahidol University, Bangkok 10400, Thailand; balasubramanian.bri@alumni.mahidol.ac.th (B.B.);; 2Biodiscovery Institute, University of Nottingham, Nottingham NG7 2RD, UK; 3Graduate Program in Molecular Medicine, Faculty of Science, Mahidol University, Bangkok 10400, Thailand; 4Oncology Unit, Department of Medicine, Sunpasitthiprasong Hospital, Ubon Ratchathani 34000, Thailand; 5General Surgery Division, Department of Surgery, Sanpasitthiprasong Hospital, Ubon Ratchathani 34000, Thailand; 6College of Medicine and Public Health, Ubon Ratchathani University, Ubon Ratchathani 34190, Thailand; 7Department of Pathology, Rajavithi Hospital, Bangkok 10400, Thailand; 8Department of Anatomy, Faculty of Science, Mahidol University, Bangkok 10400, Thailand

**Keywords:** FGFR inhibitors, cholangiocarcinoma, biliary tract cancer, molecular targeted therapy, therapeutic biomarker, precision medicine

## Abstract

**Simple Summary:**

Cholangiocarcinoma (CCA) is an architecturally complex and highly heterogeneous tumour and is difficult to diagnose at early stages due to its late onset and asymptomatic nature. Recent studies have focused on the Fibroblast Growth Factor Receptors (FGFRs), a sub-family of Tyrosine Kinase Receptors (RTKs), as promising targets for therapy. Pemigatinib, a small-molecule inhibitor of FGFR, was the first FDA-approved targeted therapy drug for CCA patients with FGFR2 fusions who had failed first-line chemotherapy. However, only a limited cohort of patients benefit from this therapeutic strategy as the mutation does not necessarily correlate to the response. In this study, aberrant FGFR expression in CCA samples was found using a bioinformatics approach and further confirmed by immunohistochemistry. PD173074, a selective pan-FGFR inhibitor, was found to be sensitive to CCA cell lines with FGFR expression, suggesting that it can be used to suppress CCA cells even without the FGFR2 fusions. Moreover, correlation analysis of publicly available cohorts suggested the possibility of crosstalk amongst the FGFR and EGFR family of receptors and dual inhibition of PD173074 and erlotinib was found to be synergistic in CCA using cell lines and patient-derived complex models. Thus, this study suggests further clinical investigation of PD173074, as well as other FGFR inhibitors, to benefit a larger cohort of CCA patients and novel therapeutic strategies involving dual inhibition of FGFRs and EGFR.

**Abstract:**

Cholangiocarcinoma (CCA) is an architecturally complex tumour with high heterogeneity. Discovery at later stages makes treatment challenging. However, the lack of early detection methodologies and the asymptomatic nature of CCA make early diagnosis more difficult. Recent studies revealed the fusions in Fibroblast Growth Factor Receptors (FGFRs), a sub-family of RTKs, as promising targets for targeted therapy for CCA. Particularly, FGFR2 fusions have been of particular interest, as translocations have been found in approximately 13% of CCA patients. Pursuing this, Pemigatinib, a small-molecule inhibitor of FGFR, became the first targeted therapy drug to be granted accelerated approval by the FDA for treating CCA patients harbouring FGFR2 fusions who have failed first-line chemotherapy. However, despite the availability of Pemigatinib, a very limited group of patients benefit from this treatment. Moreover, as the underlying mechanism of FGFR signalling is poorly elucidated in CCA, therapeutic inhibitors designed to inhibit this pathway are prone to primary and acquired resistance, as witnessed amongst other Tyrosine Kinase Inhibitors (TKIs). While acknowledging the limited cohort that benefits from FGFR inhibitors, and the poorly elucidated mechanism of the FGFR pathway, we sought to characterise the potential of FGFR inhibitors in CCA patients without FGFR2 fusions. Here we demonstrate aberrant FGFR expression in CCA samples using bioinformatics and further confirm phosphorylated-FGFR expression in paraffinised CCA tissues using immunohistochemistry. Our results highlight p-FGFR as a biomarker to guide FGFR-targeted therapies. Furthermore, CCA cell lines with FGFR expression were sensitive to a selective pan-FGFR inhibitor, PD173074, suggesting that this drug can be used to suppress CCA cells irrespective of the FGFR2 fusions. Finally, the correlation analysis utilising publicly available cohorts suggested the possibility of crosstalk amongst the FGFR and EGFR family of receptors as they are significantly co-expressed. Accordingly, dual inhibition of FGFRs and EGFR by PD173074 and EGFR inhibitor erlotinib was synergistic in CCA. Hence, the findings from this study provide support for further clinical investigation of PD173074, as well as other FGFR inhibitors, to benefit a larger cohort of patients. Altogether, this study shows for the first time the potential of FGFRs and the importance of dual inhibition as a novel therapeutic strategy in CCA.

## 1. Introduction

Cholangiocarcinoma (CCA) is a group of malignancies of the biliary tract tree with a heterogenous nature and complex tumour microenvironment. It constitutes approximately 15% of all primary liver cancers and 3% of all gastrointestinal malignancies [1,2,3]. The socioeconomic burden of CCA is severe, particularly in low- and middle-income countries with high prevalence, such as Thailand. CCA is almost 100 times more prevalent in Thailand (85 per 100,000) than the Western countries (0.8–4 in 100,000) [2]. Due to its late onset and asymptomatic nature, CCA is an accumulation of decade-long progression of the oncogenic process. As a result, it is usually discovered in later stages when the cancer is substantially progressed due to its asymptomatic nature and the lack of appropriate early detection methods. This makes treatment challenging. Moreover, patients are generally minimally responsive to systemic chemotherapy. Surgery with curative intent, is technically challenging and is accompanied by a lengthy recovery. Yet, it does not improve overall patient survival, particularly in the intrahepatic CCA subtype [4]. Therefore, there is an urgent need for accurate patient stratification, as evidenced by the increasing studies which support the use of biomarker-guided treatment for the CCA [5]. The paucity of alternative options, such as targeted therapy, limits the practicality of precision medicine. However, fibroblast growth factor receptor (FGFR) signalling has emerged as a promising actionable target in CCA following the accelerated approval of FGFR inhibitor, pemigatinib, for patients harbouring FGFR2 fusions and have failed first-line chemotherapy [6].

Fibroblast Growth Factors (FGFs) and their related receptors (FGFRs) are involved in several physiological processes, which makes them susceptible to dysregulation by cancer cells. There is substantial evidence implicating the involvement of FGFR signalling in the pathogenesis of various cancer types; the mechanism, however, is tumour-specific [7]. In CCA, FGFR2 fusions are presented in approximately 13% of all cases, almost exclusively in the intrahepatic CCA [8]. However, the understanding of FGFR inhibitors in CCA is limited, and as a result, several complications have arisen, the most prominent being acquired resistance to prolonged exposure to FGFR inhibitors. Most patients develop secondary mutations, such as gatekeeper mutations in the kinase domains, as a mechanism of acquired resistance to FGFR inhibitors [9]. Similarly, the emergence of polyclonal mutations has resulted in acquired resistance to pemigatinib as well. Other modes of resistance to FGFR inhibitors include the activation of alternative signalling pathways, particularly other membrane’s RTK signalling [10]. Hence, the mechanisms of inhibition and ways to overcome resistance to FGFR inhibitors remain to be fully characterised in CCA. Furthermore, the role of FGFR inhibition in CCA patients without FGFR2 fusion is poorly understood. There are several FGFR inhibitors currently in various stages of clinical trials in CCA. However, they are exclusive for patients with FGFR2 fusions [NCT04919642, NCT04093362, NCT04526106, NCT03656536]. Therefore, the aim of this study is to further the understanding of FGFR expression and its inhibition in CCA beyond fusions. The results from our study may facilitate the use of FGFR inhibitors in a larger population of CCA patients, irrespective of FGFR2 fusions. Previously, we established FGFR1–4 are likely to be actionable targets and that they are often co-expressed with the EGFR family in some CCA subtypes [11]. Here, we utilised a combination of computational and pre-clinical experimentation to elucidate the role of FGFs and their related receptors in CCA.

## 2. Materials and Methods

### 2.1. Data Acquisition and Pre-Processing

Baseline mRNA expression of the FGF (*FGF1–23*) and FGFR (*FGFR1–4*) family in CCA tumours and adjacent noncancerous tissues were analysed using the transcriptomic profiles of patient-derived CCA tumours were collated from 10 independent yet related datasets from Gene Expression Omnibus (GEO; accessed on 20 June 2022) (GSE132305, GSE22633, GSE26566, GSE32225, GSE32879, GSE35306, GSE57555, GSE66255, GSE76279 and GSE89749) in a previous study [11]. For validation sets, we applied RNA-seq data obtained from GEO using the search terms “cholangiocarcinoma” AND “human” AND “RNA-seq”. Results were filtered to select only processed RNA-seq data, resulting in the reads per kilobase million (RPKM) counts matrix from GSE107943, containing 30 CCA samples with 27 matched normal liver tissues. For the second cohort, RNA-seq data of TCGA-CHOL was acquired from cBioportal (www.cbioportal.org; accessed on 20 June 2022), containing 36 CCA tumour samples. Each dataset was treated independently. Differential Gene Expression (DGE) analysis was performed using the ‘limma’ and ‘deseq2′ package, version 4.0.2 in ‘R’. Unsupervised clustering analysis was performed using the ‘pheatmap’ package, version 4.0.2 in ‘R’.

### 2.2. Enrichment Analysis

Pathway Enrichment analysis for the DEGs obtained was performed using the web tool ‘Enrichr’ [12,13,14]. The resultant terms were filtered for statistical significance (*p* < 0.05) and ordered according to the combined score. The top enriched results from Bioplanet, KEGG, MSigDB and WikiPathways databases was represented. Enrichment of small molecule inhibitors that target FGFR1–4 was performed using Enrichr with the search terms “FGFR1”, “FGFR2”, “FGFR3”, and “FGFR4”. Publicly available kinase inhibitor screening database, HMS LINCS KinomeScan (https://lincs.hms.harvard.edu/db/; accessed on 2 July 2022). Statistically significant kinase inhibitors (*p* < 0.05) were ordered according to combined score, also known as enrichment score and the top 10 results are reported.

### 2.3. Survival Analysis

Survival analysis was performed using the ‘survival’, ‘survminer’, and ‘survplot’ packages in R version 4.0.2. The Cox-Proportional Univariate Regression model was applied to each gene in the FGFR signature (FGFR1–4), and Hazard Ratios (HR) and Wald Statistic *p*-value were tabulated and plotted for the validation datasets. Kaplan–Meier plots for high and low expression groups of p-FGFRs are plotted using Graphpad Prism 9.

### 2.4. Immunohistochemistry

Formalin-fixed-paraffin-embedded (FFPE) tumour samples were retrieved from the Department of Pathology, Ramathibodi Hospital, Mahidol University and the Department of Pathology, Rajavithi Hospital, Bangkok, Thailand, for immunohistochemistry assessment. The tissues were histologically confirmed as mass-forming CCA by the pathologists of the institutes, respectively. Serial 5-µm-thick sections of FFPE tissue were cut for preparation of tissue microarray (TMA) by a pathologist. Each case was tested using the following primary antibodies: anti-FGFR3 and anti-phospho-FGFR antibodies. The staining was visualised using DAB (3,3′-diaminobenzidine) substrate conjugated with HRP. The signal intensity of DAB was measured and quantified using an integrated protocol on Image J [15].

### 2.5. Cell Line Culture

A total of six different patient-derived CCA cell lines, namely HuCCA-1, KKU-100, KKU-213, RBE, and TFK-1, were used in this study. HuCCA-1, KKU-100, and KKU-213 were purchased from the Japanese Collection of Research Bioresources Cell Bank. RBE and TFK-1 were gifted from Prof. David Bates, University of Nottingham, UK. The cell lines were maintained in a humidified 5% CO_2_ incubator at 37 °C in RPMI medium 1640 supplemented with 10% foetal bovine serum and 1× Antibiotic-Antimycotic.

### 2.6. Cell Viability Assays

Cells were seeded in 96-well plates at 5000 cells per well density for 24 h. The cells were treated with FGFR inhibitors at varying concentrations (0.1, 1, 10 and 100 mM) for 72 h. Cell viability was measured every 24 h. Cells were incubated with MTT (final concentration 0.5 mg/mL) in a complete medium for 2 h. The crystals were dissolved in DMSO and measured at OD 540 nm.

### 2.7. Colony Formation Assays

Cells were seeded in a 6-well plate at 1000 cell density overnight. The next day the cells were treated with varying concentrations of FGFR inhibitor in a 10% FBS-supplemented medium (3 mL). The cells were grown for 14 days at 37 °C in a humidified 5% CO_2_ incubator. Then the cells were washed with warm PBS once before fixing with Acetic Acid/Methanol (1:7 *v*/*v*) for 5 min. Then, the fixing solution was removed, and cells were washed with PBS one time, stained with 0.5% Crystal Violet (25% Methanol) for 30 min, followed by washing with water.

### 2.8. Quantitative Real-Time Polymerase Chain Reaction (qRT-PCR)

Total mRNA extraction was performed using Total RNA Mini Kit (Blood/Cultured cells) (Geneaid, New Taipei City, Taiwan) and reverse transcribed using ImProm-II™ Reverse Transcription System (Promega, Madison, WI, USA). qRT-PCR was performed using Faststart universal SYBR green Master (Roche, Mannheim, Germany). Specific primers to FGFR1, FGFR2, FGFR3 and FGFR4 were designed to target the common region in existing isoforms of the gene. 18S mRNA was used as an internal control gene.

### 2.9. Western Blotting Analysis

Protein lysates from cell lines are collected at 80% confluence in NP-40 lysis buffer containing 150 mM Tris-HCl pH 7.4, 150 mM NaCl, 5 mM EGTA, 5 mM EDTA, 0.1% SDS, 1% sodium deoxycholate, 1% Nonidet P-40, 1× protease inhibitor cocktail (Roche), 50 mM NaF, 2 mM Na_3_VO_4_, 40 mM β-glycerophosphate and 1 mM dithiothreitol. Protein concentration was estimated using Bradford assay (Bio-Rad, Hercules, CA, USA), and 30 µg of proteins were separated using 10% SDS-PAGE and transferred onto a nitrocellulose membrane. Non-specific proteins were blocked by 3% BSA in TBS-T. The membrane was incubated with primary antibodies: goat anti-pFGFR (3471), rabbit anti-FGFR1 (D8E4 #9740), rabbit anti-FGFR2 (D4H9 #11835), rabbit anti-FGFR3 (C51F2 #4574) and rabbit anti-FGFR4 (D3B12 #8562), rabbit anti-phospho STAT3 (D3A7 #9145) and mouse anti-STAT3 (610189) followed by goat anti-rabbit (sc-2004) horseradish peroxidase (HRP)-conjugated secondary antibodies (Santa Cruz Biotechnology (Santa Cruz, CA, USA)). Goat anti-GAPDH (sc-48166) (Santa Cruz Biotechnology) primary antibodies were used for the normalisation of gel loading. Clarity Western ECL reagent (Bio-Rad) was utilised to detect chemiluminescent signals, which were visualised by the G-Box Chemi XL system (Syngene, Cambridge, UK).

### 2.10. Human Phospho-Protein Proteome Profiler Array

Lysates (400 mg) from DMSO (vehicle) and PD174074 treated cells were incubated with the Human Phospho-Kinase Array Kit (R&D Systems, Minneapolis, MN, USA) according to the manufacturer’s instructions. A chemiluminescent signal was acquired using Western ECL reagent (Bio-Rad) and G-Box Chemi XL system (Syngene, Cambridge, UK). The signal intensities were analysed using the Image Lab software (Bio-Rad).

### 2.11. Caspase 3 Activity Assay

The caspase-3 activity in CCA cells was measured by the Caspase-3 Assay Kit (ab39401, Abcam (Minneapolis, MN, USA)). The CCA cells were treated with DMSO (vehicle) and PD173074 for 24 h; cell lysates were harvested on ice. Protein was estimated using Bradford’s reagent (Bio-Rad); concentration was adjusted to 100 mg/well and was assayed using the kit. Caspase activity was measured at 400 nm using a TECAN SPARK^®^ microplate reader.

### 2.12. Flow Cytometry Apoptosis Assay

CCA cells were seeded in a 6-well plate at 3 × 10^5^ per well for 24 h. The next day the medium was removed and treated with 5 and 10 µM of a compound or DMSO 0.001% as vehicle control for 24 h in a 10% FBS-supplemented medium. Then, the cells were collected by trypsinisation, washed with PBS and resuspended in Annexin V-binding buffer. Then the cells were stained with Annexin-V (0.3 µg/mL) and PI (2 µg/mL) for 5 min at RT in the dark. FACS analysis was performed with Attune NxT Flow Cytometer (Thermo Fisher Scientific, Waltham, MA, USA). The percentage of apoptotic cells is an addition to the percentage of early and late apoptotic cells.

### 2.13. Synergy

The cells were seeded in 96-well plates at 2500 per well density overnight. The next day the cells were treated with drug combinations as a matrix for 72 h. MTT assay was performed as mentioned earlier. The synergy score was calculated using Loewe Additivity Model, and synergy maps were plotted using the ‘synergyfinder’ package in R [16].

### 2.14. 3-D Tumour Growth Assay

Primary cells were resuspended in ice-cold Cultrex basement membrane extract (BME) (9 mg/mL: Trevigen (Minneapolis, MN, USA)) diluted in modified RPMI-1640 (Life Technologies (Carlsbad, CA, USA); phenol red free with 6 mmol/L D-Glucose and pH 6.8) and plated at 6250 tumour cells with Mesenchymal cells (bone marrow-derived) (ScienCell) into low adherent, black-walled, clear bottom, 384 well plates. Drugs were serially diluted in modified RPMI-1640, and 13 µL of the drug was added in six replicates on day 3. Drugs used in combination were premixed and serially diluted together before adding to the assay. Drug exposure was for 96 h before the final endpoint readings. The AlamarBlue assay (Invitrogen (Waltham, MA, USA); 10% (*v*/*v*). 37 °C for 1 h) was used to monitor cell growth daily using a fluorescent plate reader (FLUOstar Omega, BMG Labtech (Cary, NC, USA)). Drug sensitivity was calculated as a percentage of matched untreated control, and IC50curves were determined using GraphPad Prism 5 (GraphPad Software Inc. (San Diego, CA, USA), nonlinear curve fit of Y = 100/(1 + 10 ((Log1C50-X) × HillSlope).

## 3. Results

### 3.1. Aberrant Expression of FGFRs Was Observed in CCA Tissues Using an Integrated Bioinformatics Approach

The Cancer Genome Atlas (TCGA) is a pan-cancer database containing RNA sequencing data of tumours and normal adjacent tissues. We observed that FGFR mRNA expression was elevated almost exclusively in CCA tissues when compared to normal adjacent tissues than other leading cancers (breast, lung, liver, prostate) (Appendix A). Hence, we hypothesised that dysregulation of the FGF pathway may play a role in CCA pathogenesis. To explore the gene expression profile of the FGF family in CCA, we compared the mRNA expression levels of FGF ligands and their receptors in CCA (*n* = 704) and normal (*n* = 165) tissues from our previously collated expression data pooled from 10 independent microarray studies [11]. We found that all four of the functional FGFRs (*FGFR1*–*FGFR4*) were significantly upregulated in CCA (*p* < 0.001; *p* = 0.0008, *p* < 0.001, *p* = 0.0023) (Figure 1A–D). In addition, we also observed aberrant expression of several FGF ligands in the CCA tissues. *FGF1*, *FGF3*, *FGF12*, and *FGF20* were significantly upregulated (*p* = 0.0377; *p* = 0.0371; *p* = 0.0487; *p* = 0.0088) (Figure 1E–G), whereas *FGF4*, *FGF7*, *FGF8*, *FGF11*, *FGF18*, and *FGF22* were significantly downregulated in CCA tissues when compared to normal (*p* < 0.0001) (Appendix A). We also discovered that FGF ligands were not only upregulated in CCA tissues, but they were also positively correlated with the expression of FGFRs. *FGF1* and *FGF3* were significantly positively correlated with receptors FGFR1 (*r* = 0.59, *p* < 0.0001; *r* = 0.49 and *p* < 0.0001) *FGFR2* (*r* = 0.49, *p* < 0.0001; *r* = 0.32, *p* < 0.0001), while *FGF20* was significantly positively correlated with *FGFR1* (*r* = 0.28, *p* < 0.0001) and *FGFR3* (*r* = 0.36, *p* < 0.0001). A correlation matrix summarises these results (Appendix A). Altogether, these results confirm the aberrant expression of FGFs and FGFRs, therefore, implicating their role in CCA pathogenesis.

### 3.2. High Expression of FGFRs Is Associated with Cancer Hallmark Pathways in CCA

Using hierarchical clustering analysis, we evaluated the overall gene expression profile of FGFRs in CCA tissues, as illustrated in the heatmap (Figure 1H). We further explored the integrated gene signature of these groups using multi-dimensional scaling (MDS) analysis, which illustrated that each group had a distinct overall molecular profile at a transcriptome level that can be defined by their FGFR expression (Figure 1I). We then sought to identify the underlying transcriptomic signatures of the high-expression group by comparing it against the group with low expression of FGFRs. This resulted in a total of 197 upregulated genes and 173 downregulated genes in the high-expression group, as illustrated by the volcano plot (Figure 1J). Statistically significant DEGs (Appendix A) were selected for pathway enrichment analysis. Signalling pathways related to growth factors of RTKs, such as EGFR, PDGFR, and VEGFR family, were significantly enriched. Similarly, pathways related to adhesion, migration, and invasion, such as signalling by focal adhesion kinases and regulation of extracellular matrix, were also enriched. Moreover, other cancer hallmark pathways, i.e., angiogenesis, apoptosis and p53 pathways, are also significantly enriched. Intracellular signalling cascade pathways, such as PI3K-Akt and JAK-STAT signalling pathways, are also enriched (Figure 1K). Altogether, these findings reiterate the involvement of FGFRs in CCA oncogenesis and therefore highlight their importance as potential targets for therapy.

### 3.3. Clinical Relevance of FGFR Gene Signature Suggests That FGFR Expression Can Be Used to Stratify CCA Patients According to Risk

We evaluated the clinical relevance of FGFR gene expression by utilising two independent cohorts, GSE107943, a dataset containing CCA tissues (*n* = 30), as well as normal adjacent tissues (*n* = 27), and the TCGA-CHOL cohort containing 36 CCA tissues and nine adjacent non-cancerous tissues. In both cohorts, we found that FGFR expression is significantly elevated in CCA when compared with the normal adjacent tissues (Figure 2a,d). Moreover, using Kaplan–Meier survival analysis, we found that higher FGFR expression predicted a better clinical outcome. The results showed that a higher FGFR expression average resulted in a significantly better overall survival (OS) (*p* = 0.005) and disease-free survival (RFS) (*p* = 0.001) in the GSE107943 cohort (Figure 2b,c). However, the results from the TCGA cohort were inconclusive as they were not statistically significant (Figure 2e,f). Since ectopic expression of FGFRs can lead to constitutive activation of FGFRs and subsequent signalling pathways, we investigated the expression of phosphorylated FGFR in CCA. Although there are four individual receptors, the intracellular tyrosine kinase domain is similar in all the FGFRs; amongst the catalytic domain, the tyrosine residues, Tyr653 and Tyr654, are the most important for activation of all the FGFRs and the subsequent signalling cascades [17]. Surgically resected CCA tissues were obtained from Ramathibothi, Bangkok, Thailand and stained with phosphorylated-FGFR (p-FGFR) antibody and visualised under 20× magnification (Figure 2g). p-FGFR staining was observed in the membrane, cytoplasm and the nucleus; the intensity of the staining was classified into low and high intensity based on median-cut-off of the intensity scores (Figure 2h). The correlation between the expression of p-FGFR and patient survival was determined using the Kaplan–Meier survival plot (Figure 2i). There was no significant correlation between the expression of p-FGFR and patients’ survival time. However, the trend suggests that high expression of p-FGFR is associated with poorer overall survival. Furthermore, p-FGFR expression is associated with over 1-year survival (*p* = 0.0177) {Table 1}. Altogether, these results suggested that FGFR expression might be a useful prognostic marker for predicting a better clinical outcome. In addition, these results demonstrate that pFGFR is a biomarker for constitutive activation of FGFR signalling in CCA and for targeted therapy. This further confirms that active FGFRs might be actionable targets in CCA treatment.

### 3.4. PD173074 Is a Potential Candidate for FGFR Inhibition in CCA

As previous results illustrated the importance of activated FGFR signalling as potentially actionable targets in CCA, we found that PD173074 was enriched to selectively target all the FGFRs with the highest combined score in Enrichr (Figure 3a). Hence, PD173074 will be used henceforth to study FGFR inhibition in this study. The Cancer Cell Line Encyclopaedia (CCLE) is a publicly available database that contains RNA sequencing data for many cell lines of different cancer types, including 24 CCA cell lines. This database was used to explore the baseline mRNA expression and mutation status of the four FGF receptors, *FGFR1*, *FGFR2*, *FGFR3* and *FGFR4*, in the CCA cell lines. The expression levels and the mutation status of the 24 CCA cell lines are visualised as a oncoplot (Figure 3b). The heatmap shows the expression levels, and the oncoplot indicates mutations in each receptor. While none of the cell lines harbours *FGFR2* fusions, HUCCT-1 and SNU245 have missense mutations in the *FGFR2* gene. Likewise, KKU-100 and TFK-1 cells have missense mutations in the *FGFR3* gene (Appendix A). No other mutations in the FGFRs were present in any of the other CCA cell lines. However, ectopic expression of the FGFRs was present in many of the cell lines. KKU-100, KKU-213, RBE and TFK-1 cell lines were chosen as representative cell lines for this study (Appendix A). KKU-100 and TFK-1 are established from extrahepatic CCA tumours, whereas KKU-213 and RBE were established from intrahepatic CCA tumours. KKU-100 has a comparatively low expression of FGFRs and has an *FGFR3* mutation. KKU-213 has moderate expression of FGFRs but no mutations. TFK-1 also has moderate expression of FGFRs and an *FGFR3* mutation. The baseline mRNA expression of *FGFR1*, *FGFR2*, *FGFR3* and *FGFR4* was measured using qRT-PCR in the selected cell lines (Figure 3c–f). Compared to HuCCA-1 (the lowest FGFR-expressing cell line), the RBE cell line had the highest expression of *FGFR1*, whereas KKU-213 had the highest expression of *FGFR2*, RBE and KKU-213 had similar expression levels of *FGFR3*. TFK-1 had the highest expression of *FGFR3* and *FGFR4*. These results indicate that these cell lines are appropriate models to study FGFR inhibition in CCA and explore its potential as a therapeutic strategy.

### 3.5. Sensitivity to PD173074 in CCA Cell Lines

PD173074 effectively reduced cell viability in a dose-dependent manner in all the CCA cell lines except KKU-100, which only responded to high concentrations (Figure 3g–j). TFK-1, KKU-213 and RBE cells responded to PD173074 treatment with the IC50 of ~6.6 µM, ~8.4 µM, and ~11 µM, respectively (Figure 3k). KKU-100, with low mRNA expression of FGFR, responded to a slightly higher dose of PD173074 with an IC50 of ~16 µM. As FGFR mutations, apart from *FGFR2* fusions, are rare in CCA, KKU-213 and RBE cell lines were selected for further analysis. Despite having baseline mRNA expression of the receptors, both cell lines do not harbour FGFR mutations, and considering their sensitivity to FGFR inhibition, it is likely that they will better represent a larger cohort of CCA patients. KKU-213 and RBE were treated with PD173074, and we found that colony formation was inhibited in both cell lines (Figure 3l). Altogether, this shows that PD173074 is cytotoxic as it inhibits cell proliferation, colony formation, and cell survival in CCA. Both the cell lines were sensitive to PD173074 in combination with Gemcitabine and Cisplatin (Appendix A). Therefore, PD173074 is a promising novel therapeutic for patients with elevated FGFR expression in CCA.

### 3.6. PD173074 Induces Apoptosis in CCA

FGFR inhibitor PD173074 significantly increased the number of apoptotic cells by ~15% in KKU-213 and ~10% in RBE cells at 5 00B5M concentration (Figure 4a–c). In KKU-213 cells, treatment with a higher concentration of PD173074 (10 µM) also had a similar effect, whereas in RBE cell lines, there was only a slight increase in apoptotic cells compared to vehicle control, but it was not statistically significant. In addition, caspase 3 activity was significantly increased in a dose and time-dependent manner in KKU-213 cells (Figure 4d,f). In RBE, the caspase 3 activity was significantly increased after treatment with PD173074 at 5 µM. However, the increase in activity was not as profound with 10 µM concentration, even though there was a significant increase in caspase 3 activity in a time-dependent manner (Figure 4e,g). Altogether, these results demonstrate that PD173074 induces apoptosis in CCA.

### 3.7. PD173074 Blocks FGF-Stimulated FGFRs and Further Downstream Signalling

To confirm whether PD173074 does, in fact, inhibit the activation of FGFRs, the effect of the inhibitor on the phosphorylation of FGFR was investigated in KKU-213 cells. KKU-213 cell line was more responsive to FGFR inhibition (lower IC50) than RBE. The receptors were when stimulated with 40 ng/µL of FGF-1, and we found constitutive activation of FGFRs in the KKU-213 cell line; however, stimulation with FGF-1 slightly increased the levels of p-FGFR. Treatment with PD173074 effectively reduced the phosphorylation of constitutively active FGFR, and FGF-1 were not able to stimulate the blocked receptors. In fact, the inhibitor was more effective in reducing the p-FGFR in the presence of FGF-1 (Figure 5a). Altogether, this result confirms that PD173074 blocks p-FGFR in CCA. FGFR inhibition works even better in the presence of FGFs, suggesting that this drug may even be effective in the presence of growth factors from the tumour microenvironment. Moreover, activated FGFR (phosphorylated FGFR) can trigger cellular signalling via various cell signalling pathways, such as RAS/MAPK, PI3K/AKT, JAK/STAT and PLCγ signalling cascades. Hence, we tested which downstream signalling pathways are affected using a kinase antibody array (Figure 5b). Interestingly, some of the kinases were downregulated upon treatment, whereas there were several kinases with increased phosphorylation levels post-treatment (Figure 5c,d). Phospho-p53 was the most inhibited by PD173074 treatment, considering that KKU-213 has a TP53 mutation (p.V31I); this suggests that this inhibitor can be effective even in the presence of p53 mutations. Additionally, downstream effector kinases, such as STAT3 and PLCG1, were also reduced after treatment, suggesting that in CCA, the downstream of FGFR signalling is via the STAT3 pathway. This finding is confirmed by western blotting in KKU-213 and another cell line, RBE (Figure 5g,h and Appendix A). Interestingly, multiple kinases were upregulated following treatment, indicating that inhibition of the FGFR signalling eventually results in activation of another signalling. To identify the functional role of these networks, the upregulated kinases were input into the web tool StringDB to (a) identify the networks and protein-protein interaction (PPI) of these kinases and (b) pathways that are related to these networks. StringDB collates information about proteins and their interactions using experimental and theoretical knowledge to build PPI networks. From this analysis, it is evident that RTK signalling, particularly FGFR signalling in disease, was inhibited by PD173074. However, the kinases upregulated were related to EGFR and other intracellular signalling pathways (Figure 5e,f).

### 3.8. Combination Treatment with Erlotinib Increases the Sensitivity of CCA Cells to Inhibition by PD173074

The previous findings showed the possibility of a compensatory mechanism for FGFR inhibition by PD173074 by upregulation of kinases from the EGFR pathway in CCA cells. Hence, we hypothesised that combining PD173074 with EGFR inhibitors will have a synergistic effect on CCA. We tested the efficacy of FGFR inhibition in KKU-213 and RBE cells in the presence of EGFR inhibitor erlotinib. Combination treatment with PD173074 and erlotinib demonstrated synergistic effects in both KKU-213 and RBE cancer cell lines. This dose-dependent response is visualised as a dose–response matrix (Figure 6a,b). The effect of single inhibition with PD173074 and dual inhibition with erlotinib on cell viability is visually represented using a dose–response curve (Figure 6c,d) in both the cell lines tested. Synergy was evaluated using three different mathematical models: Loewe’s additivity (Figure 6e,h), Highest Single Agent (HSA) (Figure 6f,i) and Bliss independence (Figure 6g,j). The highest synergy score according to Loewe’s additivity model was 26.35 in KKU-213 cells at 0.625 µM of erlotinib and 2.5 µM of PD173074, and 13.62 in RBE cells at 0.625 µM of erlotinib and PD173074. The HSA model yielded the highest synergy score of 40.0 in KKU-213 cells at 5 µM of both PD173074 and erlotinib and 19.81 in RBE cells at 5 µM of PD173074 and 2.5 µM of erlotinib. The bliss independence model showed the highest synergy score of 23.78 in KKU-213 cells at 5 µM of both PD173074 and erlotinib and 10.32 in RBE cells at 5 µM of PD173074 and 0.625 µM of erlotinib. In addition, our results suggest that dual inhibition with PD173074 and erlotinib may offer a potential therapeutic strategy for treating CCA with a lower concentration of drugs required for maximal effect. These findings highlight the potential of combining PD173074 and erlotinib as a promising therapeutic strategies for CCA treatment.

### 3.9. Combination Treatment in a 3-D Setting with the Presence of Tumour Stromal Cells

We then evaluated this effect in primary CCA cells isolated from patients in the presence of mesenchymal stem cells (MSCs) and cancer-associated fibroblasts (CAFs) in a three-dimensional (3D) setting (Figure 7A,B). In the absence of MSCs and CAFs, the combination treatment in CCA-UK5 cells resulted in an IC_50_ of 7.397 µM, which was not different compared to a single treatment by erlotinib and PD173074 (Figure 7C) with an IC_50_ of 6.759 µM and 5.476 µM, respectively. In contrast, CCA-UK6 cells were increasingly sensitive to dual inhibition (IC50 = 1.668 µM) when compared to mono inhibition by erlotinib (IC50 = 10.38 µM) and PD173074 (IC50 = 100 µM) in the absence of MSCs and CAFs (Figure 7D). The IC50 for dual inhibition by erlotinib and PD173074 was reduced in CCA-UK5 cells in the presence of both MSCs (IC50 = 2.488 µM) and CAFs (IC50 = 2.287 µM) as opposed to alone (IC50 = 7.379 µM). A similar effect was observed in CCA-UK6 cells as well. The IC50 of the combination treatment was reduced from 1.668 µM to 0.1 µM in the presence of CAFs and increased to 8.179 µM in the presence of MSCs (Figure 7E–G). (Table 2). Together these results suggest that primary CCA cells are sensitive to dual inhibition by erlotinib and PD173074, and they are more efficacious in the presence of stromal cells, such as MSCs and CAFs, which typically drive resistance to therapeutic agents. Together these results show that FGFR and EGFR combination therapy is a promising treatment strategy for CCA.

## 4. Discussion

CCA is a highly aggressive and heterogeneous cancer of the bile duct with a dismal prognosis and complex tumour microenvironment. Increasing evidence supports the use of molecularly guided treatment in CCA to combat tumour heterogeneity [18]. As molecularly guided treatment has been increasingly used to combat the heterogeneous tumour microenvironment of CCA, targeted therapies, such as ivosidenib for patients with *IDH1* mutations and pemigatinib for patients harbouring *FGFR2* fusions, respectively, have been approved for use in CCA [19,20]. However, pemigatinib is currently only approved for use in treating advanced metastatic CCA patients that harbour *FGFR2* fusions and have failed first-line treatment. Therefore, only a small number of patients can benefit from it, as no other biomarkers indicative of treatment or response to FGFR inhibitors have been identified yet. Thus, the present study aimed to better understand the importance of FGFRs as actionable targets and the underlying mechanism of their inhibition.

To address this, we conducted an integrative transcriptomic analysis of multiple cohorts to examine the expressions of FGFR1–4 are dysregulated in CCA tissues (Figure 1A–G). Additionally, we also showed, for the first time, that p-FGFR is highly expressed in CCA tissues and is associated with poor overall survival (Figure 2). Previously, it has been established that high FGFR expression is associated with a good prognosis [21,22]. This finding is in line with our results in one of the cohorts (Figure 2e,f). While *FGFR2* fusions are commonly reported in CCA studies, mutations in the other FGFRs are infrequent. Here, we show that p-FGFR expression is the correct indicator for activated FGFR signalling, including *FGFR2* fusions. Hence it should be used as an indication for treatment with FGFR inhibitors. Apart from activating mutations and fusions, other genomic anomalies, such as gene overexpression, epigenetics, and autocrine and paracrine signalling, can also contribute to the constitutive activation of FGFR signalling pathways which consequently leads to oncogenesis [7]. This is in line with our findings as we observed elevated expression of the results from the integrated analysis illustrating the elevated expression of receptor-modulating ligands *FGF1*, *FGF3* and *FGF20* (Figure 1E–G) in CCA. In addition, *FGF1* and *FGF3* were positively correlated with *FGFR1* and *FGFR2*, while *FGF20* was positively correlated with *FGFR1*. Together, our results showed that p-FGFR expression is the correct indicator for activated FGFR signalling, which includes *FGFR2* fusions. In addition, we uncovered ligand-receptor interactions of FGF1, FGF3, and FGF20 that may drive FGFR signalling, thus providing insight into how FGFR signalling could be involved in oncogenesis.

Previously we had identified five molecular groups in a CCA cohort, in which we identified FGFR inhibitor, PD173074, to target groups with high FGFR expression [11]. This is in line with our findings; we found, using the web tool, EnrichR we identified kinase inhibitors that can specifically target all the FGFRs (Figure 3a). We found that PD173074 was significantly enriched to target the FGFRs when compared to other FGFR inhibitors. We confirmed this experimentally in in vitro studies using representative CCA cell lines, as well as near-to-patient 3D models. PD173074 as it effectively inhibited cell viability and cell survival (Figure 3g–k). Moreover, PD173074 also reduced the expression of p-FGFR in both KKU-213 and RBE cells. Moreover, PD173074 also induced apoptosis dose-dependently and increased caspase 3 activity time-dependently in CCA cells (Figure 4). Furthermore, this is the first study to report that PD173074 induces apoptosis in FGFR2 fusion-independent CCA. In addition, we also found that PD173074 is synergistic in combination with standard-of-care gemcitabine and cisplatin in CCA. Collectively, these results suggest that FGFR inhibition using PD173074 is a novel therapeutic strategy, and p-FGFR can be potentially used as a biomarker for treatment in CCA.

We have discovered that PD173074 decreases STAT3 phosphorylation, suggesting that STAT3 is downstream of FGFR signalling in CCA. STAT3 is a well-known regulator of cytokine signalling and a target for inducing apoptosis in other cancer types. Constitutive activation of STAT3 results in the dysregulation of cell cycle control, apoptosis genes, and genes that promote invasion, metastasis, and angiogenesis and lead to the suppression of host immune surveillance, all of which contribute to oncogenesis [23]. Earlier bioinformatics analysis revealed that the CCA tissues with high expression of FGFRs have genes that are associated with various RTK signalling, Focal adhesion, JAK-STAT and apoptosis-related pathways (Figure 1K). The results from the proteome array effectively reduced key players in these pathways. This opens new avenues in areas of research involving FGFRs in CCA. For instance, we found that the focal Adhesion pathway was commonly enriched in high-expression FGFR samples, and we observed that phosphorylation levels of focal adhesion kinase (p-FAK) were reduced after treatment with PD173034. This suggests that FGFRs may be associated with focal adhesion kinase pathways in CCA.

We also found that IL-6/STAT3 signalling was highly enriched in high FGFR clusters from bioinformatics analysis (Figure 1K). As FGFR inhibition reduces the phosphorylation of STAT3, it may indirectly modulate the tumour microenvironment. However, the association of FGFRs and STAT3 must be further studied and evaluated for future investigations. Our prior investigation revealed that FGFR inhibitors were enriched to reverse the signs of samples with high immune gene set expression [24]. Hence, FGFR inhibition in an immune context also needs to be further explored in CCA.

Although FGFR inhibition is a promising aspect in CCA, there is still increasing evidence of resistance to these inhibitors due to acquired secondary mutations, activation of other signalling pathways [9]. In addition, studies in other cancer types have shown interplay between FGFRs and EGFRs. For instance, a high-throughput RNA inference screen of multiple cancer cell lines identified EGFR activation could be an escape mechanism to FGFR inhibition in FGFR3 mutant cancer [25]. Furthermore, the knockdown of *ERBB3* appeared to increase the response to PD173074 in the gastrointestinal cancer [26]. These findings are consistent with our previous study [11] and the present study, indicating crosstalk amongst the EGFR and FGFR subfamilies. Moreover, our study found that combination treatment with an EGFR inhibitor, erlotinib, increased the sensitivity to PD173074 inhibition in CCA cell lines. Using near-to-patient models in a 3-D setting using primary cell lines, we observed that the combination treatment not only increases the drug sensitivity but also that it works in the presence of both CAFs and MSCs, which generally drive resistance. Therefore, the findings suggest that dual inhibition could be useful in combating the tumour microenvironment in CCA.

While the findings of this study provide valuable insights into the potential therapeutic benefits of the investigated compound, it is important to note that the study has some limitations. One such limitation is that animal work was not included in the research design. We have, however, used near-to-patient primary lines in co-culture with stromal cells for validation. Nonetheless, it is important for future studies to continue to validate the findings using in vivo models, to better understand the potential clinical implications of the compound.

## 5. Conclusions

Our research provides evidence that FGFR inhibition using PD173074 is a promising therapeutic strategy for CCA, and p-FGFR can be a useful biomarker for treatment selection and stratification. To further assess the efficacy of FGFR inhibition in CCA, further studies are needed to confirm the potential of FGFR inhibition, as well as to better understand the underlying mechanism of its inhibition. Furthermore, studies have illustrated the interplay between FGFRs and EGFRs in other cancer types, and a combination treatment with EGFR inhibitor, erlotinib, increased the sensitivity to PD173074 inhibition in CCA cell lines. Moreover, our study has shown that dual inhibition works in the presence of both CAFs and MSCs, which generally drive resistance in a 3-D setting using primary cell lines. Hence, these findings suggest that dual inhibition can be useful in combating the tumour microenvironment in CCA. Moreover, our findings should be further validated in in vivo models and in clinical trials.

## Figures and Tables

**Figure 1 cancers-15-02528-f001:**
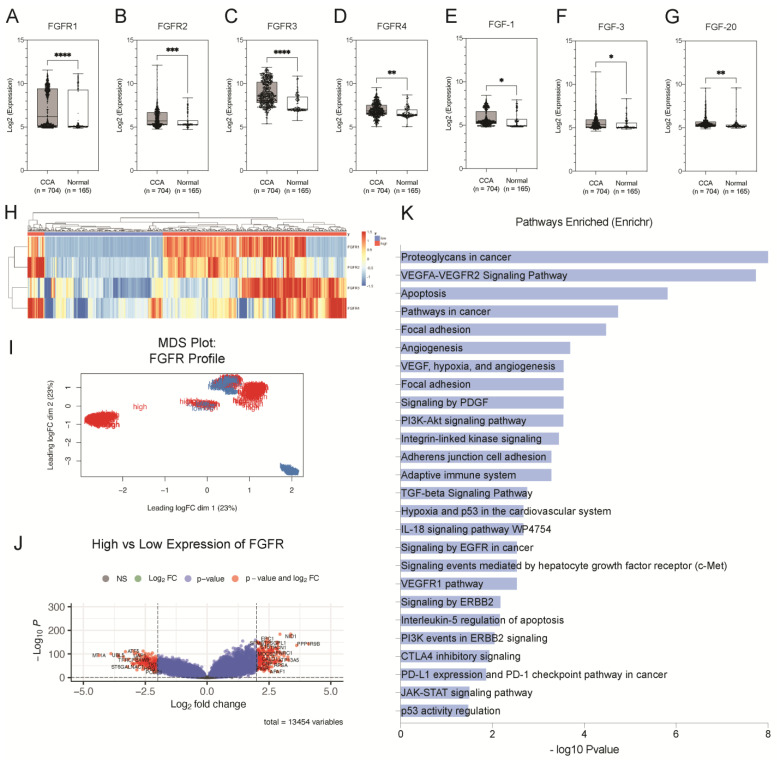
Aberrant expression of FGFRs is present in CCA tissues. Boxplot histograms represent the comparative gene expression of (**A**) FGFR1, (**B**) FGFR2, (**C**) FGFR3, (**D**) FGFR4, (**E**) FGF1, and (**F**) FGF3 and (**G**) FGF20 between CCA (*n* = 704) and normal (*n* = 165) tissues. (**H**) Heatmap represents unsupervised clustering of FGFR1, FGFR2, FGFR3 and FGFR4 expression in CCA tissues. (**I**) Multidimensional (MDS) plot of the integrated signature of the clusters using the expression profile FGFRs (*n* = 4). (**J**) Volcano plots illustrate significantly differentially expressed genes (DEGs) in the high and low expression groups as red dots (adjusted *p* < 0.001, log2 fold changes) and insignificant genes as grey. (**K**) Significantly enriched pathways of the DEGs. * *p* < 0.05, ** *p* < 0.01, *** *p* < 0.001, **** *p* < 0.0001.

**Figure 2 cancers-15-02528-f002:**
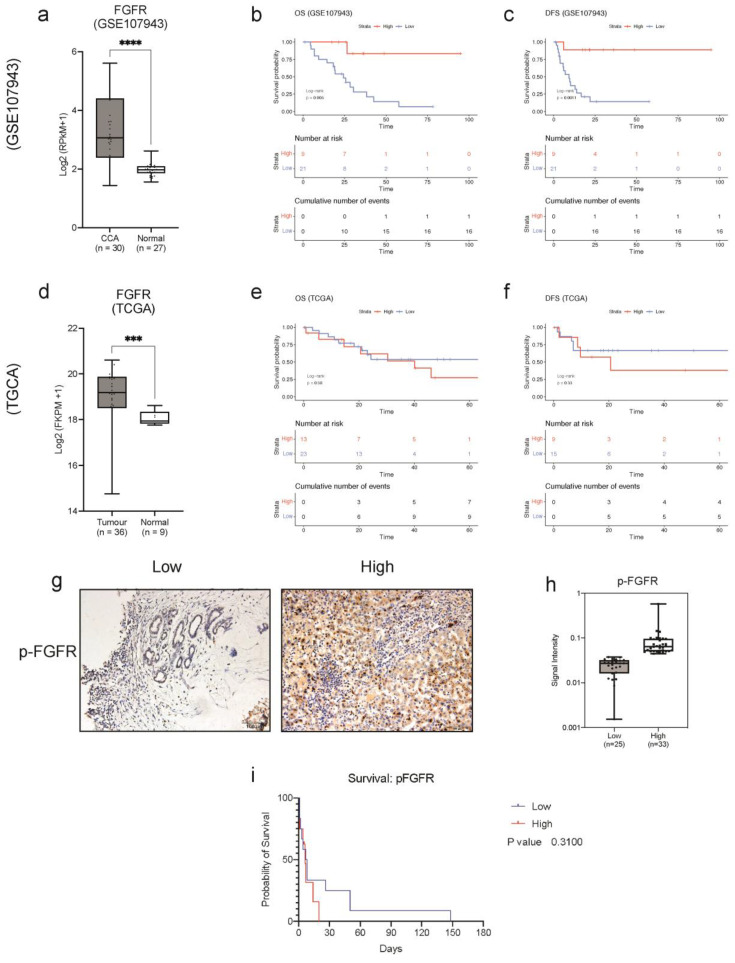
Clinical relevance of pFGFR expression in CCA. Collective mRNA expression of FGFRs (*FGFR1*, *FGFR2*, *FGFR3* and *FGFR4*) in CCA compared to normal tissues in (**a**) GSE107943 and (**d**) TCGA cohort. The Kaplan–Meier plot shows the correlation of FGFR expression with (**b**) overall survival and (**c**) disease-free survival in the GSE107943 cohort and (**e**) overall survival (OS) and (**f**) disease-free survival (DFS) in the TCGA-CHOL cohort. (**g**) Immunohistochemical staining for p-FGFR in CCA tissues (20× objective). (**h**) The boxplot shows the signal intensity values of p-FGFR for each case in the low and high expression groups. (**i**) Kaplan–Meier plot showing a correlation between patient’s survival and the expressions of p-FGFR in CCA tissues compared to each other. Survival analysis was performed using the R package ‘survival’ and ‘survminer’ and GraphPad Prism 9. *** *p* < 0.001, **** *p* < 0.0001.

**Figure 3 cancers-15-02528-f003:**
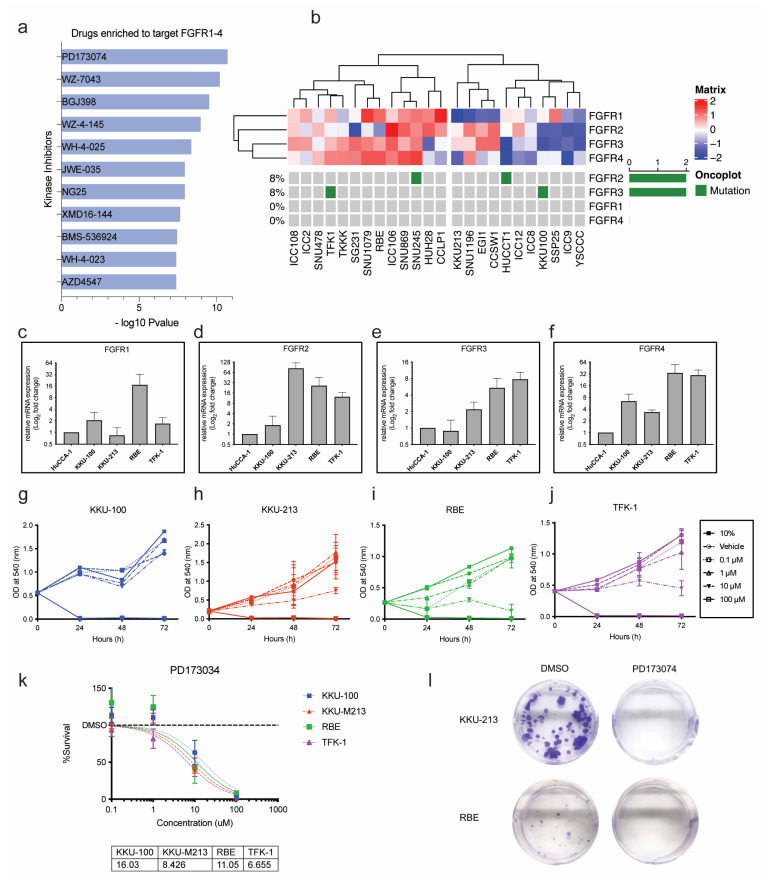
PD173074 effectively reduces cell viability and induces cytotoxicity in CCA cells. (**a**) Drugs enriched to target FGFRs in Enrichr. (**b**) Heatmap and oncoplot represent the gene expression and mutations of FGFRs in CCA cell lines from a public repository, the Cancer Dependency Map (DepMap) portal. Baseline mRNA expression of (**c**) FGFR1, (**d**) FGFR2, (**e**) FGFR3 and (**f**) FGFR4 in CCA cell lines by qPCR. CCA cells, (**g**) KKU-100, (**h**) KKU0213, (**i**) RBE and (**j**) TFK-1 were treated with varying concentrations of FGFR inhibitor, PD173074. (**k**) The cell viability results at 72 h were used to calculate the half-maximal inhibitory concentration (IC50) for each cell line. The dose–response curve was fitted to a non-linear model, and IC50 was calculated using GraphPad Prism. (**l**) The effect of PD1730374 on KKU-213 and RBE cells by colony formation assay.

**Figure 4 cancers-15-02528-f004:**
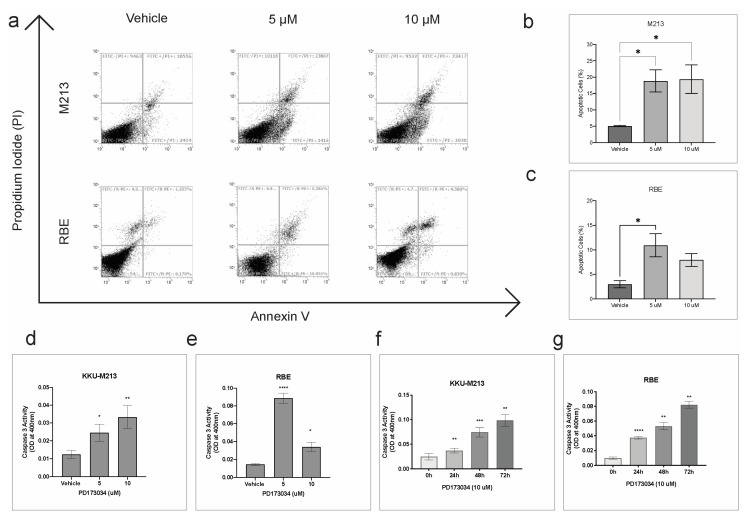
PD173034 induces apoptosis in CCA. (**a**) Flow cytometry analysis of apoptotic cells stained with Annexin V and PI post-treatment with vehicle control (DMSO), 5 and 10 µM of PD173074 for 24 h. The percentage of apoptotic cells in (**b**) KKU-213 and (**c**) RBE cells post-treatment. Caspase activity was measured as optical density. Bar graph representing optical density after treatment with PD173074 or vehicle control for 48 h in (**d**) KKU-213 and (**e**) RBE cells. Bar graph representing optical density after treatment with PD173074 time-dependently in (**f**) KKU-213 and (**g**) RBE cells. Comparisons of data between groups were made with Student’s *t*-test in GraphPad Prism. * *p* < 0.05, ** *p* < 0.01, *** *p* < 0.001, **** *p* <0.0001.

**Figure 5 cancers-15-02528-f005:**
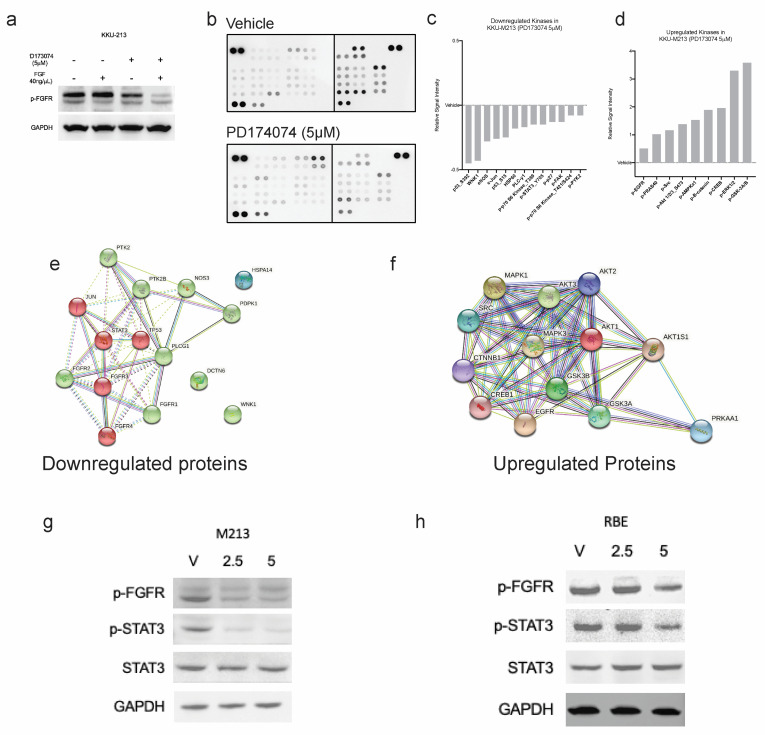
PD173074 inhibits FGF-1-activated phosphorylation of FGFRs in KKU-213 cells. (**a**) FGF-1 increases p-FGFR expression, which can be inhibited by PD173074 in KKU-213 cells. (**b**) The antibody array of phospho-kinases shows that PD173074 (5 µM) inhibits multiple kinases following 24 h treatment in KKU-213. Fold change of kinases (**c**) inhibited and (**d**) upregulated following treatment with PD173074 in KKU-213. Protein-protein interaction (PPi) network of (**e**) downregulated and (**f**) upregulated kinases. P-STAT3 expression is downregulated following p-FGFR inhibition by PD173074 in (**g**) KKU-213 and (**h**) RBE cells. (The original Western blot is in the Appendix A).

**Figure 6 cancers-15-02528-f006:**
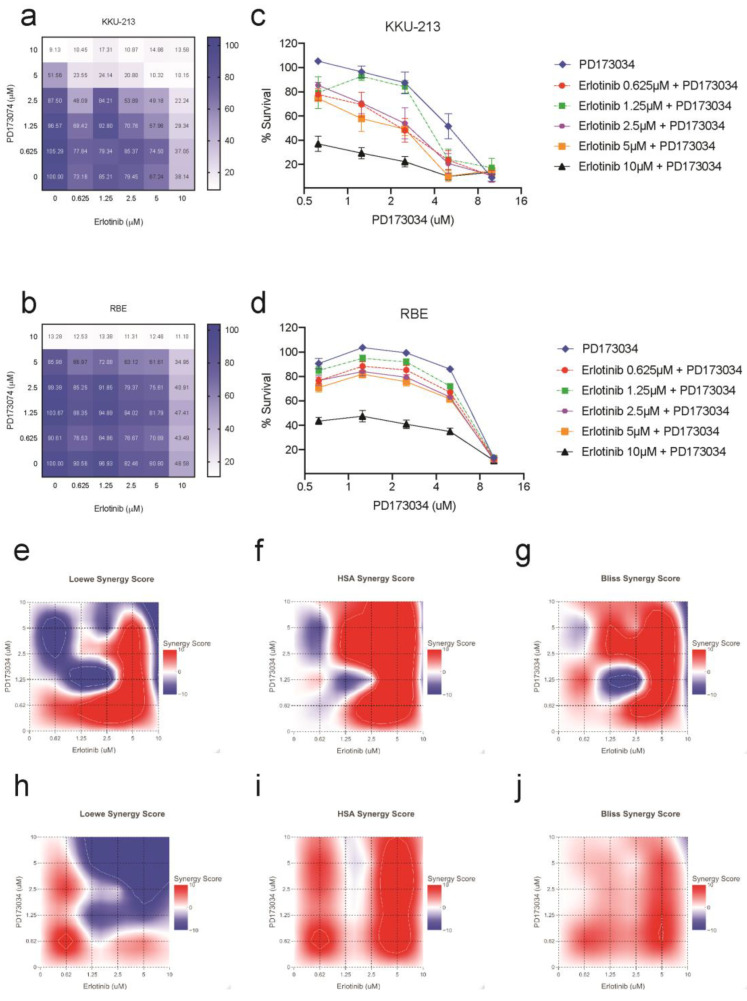
Combination treatment with PD173074 and erlotinib is synergistic in CCA. EGFR inhibitor erlotinib increases the sensitivity of PD173074 in CCA. The cells were treated with 2-fold increasing concentrations of PD173074 and Erlotinib for 96 h. The dose–response matrix shows percentage cell survival in (**a**) KKU-213 and (**b**) RBE. The dose–response curve for PD173074 in combination with increasing concentrations of erlotinib in (**c**) KKU-213 and (**d**) RBE. Synergy scores were calculated according to (**e**,**h**) Loewe’s additivity mathematical model, (**f**,**i**) Highest Single Agent (HSA) model and (**g**,**j**) Bliss Independence model using the R package ‘SynergyFinder’.

**Figure 7 cancers-15-02528-f007:**
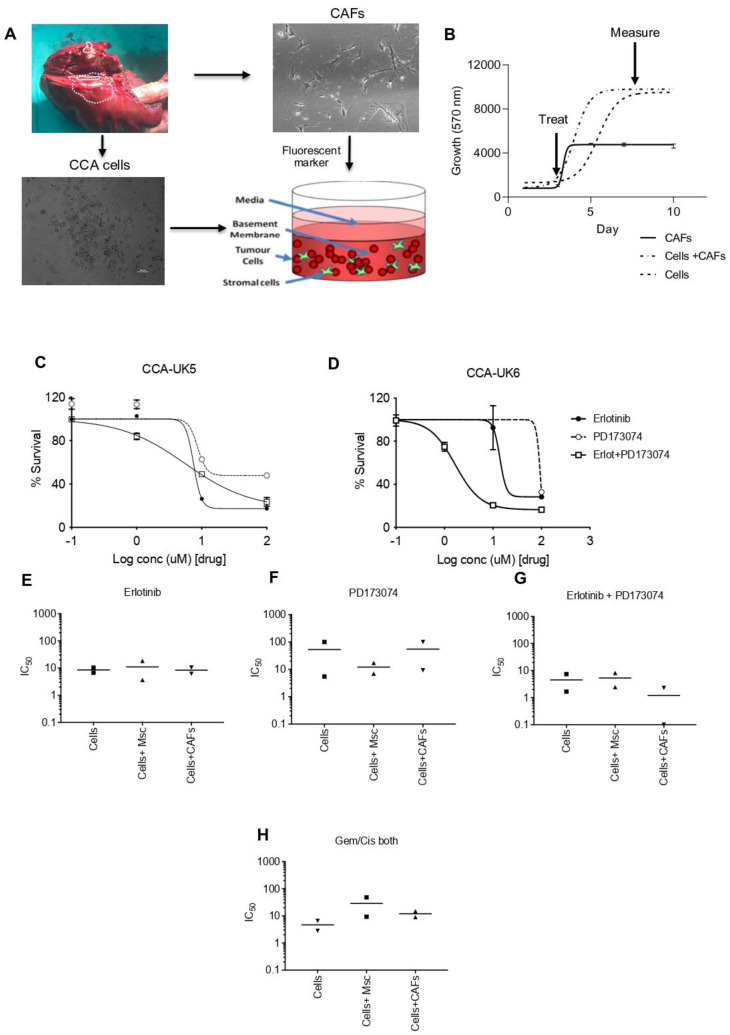
Sensitivity of PD173074 in combination treatment in patient cells derived from CCA patients in 3-D Tumour growth assays (TGA). (**A**,**B**) Two primary cell lines were derived from surgically resected CCA tissue samples CCA-UK5 and CCA-UK6 (20× magnification). Sensitivity of primary lines to (**C**,**D**) Erlotinib, PD173074 and combinations of Erlotinib and PD173074. IC50s of CCA cells, in the presence of MSCs and CAFs by (**E**) Erlotinib, (**F**) PD173074, (**G**) Erlotinib + PD173074 and (**H**) Gemcitabine/Cisplatin.

**Table 1 cancers-15-02528-t001:** Clinicopathological parameters for phosphor-FGFR expression.

	pFGFR			
Parameters	*n*	Low (*n* = 25)	High (*n* = 15)	*p*-Value
Age				
<58	7	3	4	0.4705
>58	17	12	5	0.3730
Gender				
Male	18	10	8	0.2749
Female	6	5	1	0.3508
Survival				
<1 year	18	11	7	0.5023
>1 year	6	4	2	0.0177 *

* *p* value < 0.05.

**Table 2 cancers-15-02528-t002:** IC50 values for combination treatment in patient cells derived from CCA patients in 3-D Tumour Growth Assays (TGA).

IC50 (uM)									
	Erlotinib			PD173074			Erlotinib + PD173074		
	-	+MSCs	+CAFs	-	+MSCs	+CAFs	-	+MSCs	+CAFs
**CCA5**	6.759	3.639	6.134	5.476	6.927	9.237	7.397	2.448	2.287
**CCA6**	10.38	18.41	10.51	100	17.16	100	1.668	8.179	0.1

## Data Availability

The data presented in this study are openly available in Gene Expression Omnibus (Date accessed: 2 July 2021) at GSE132305, GSE22633, GSE26566, GSE32225, GSE32879, GSE35306, GSE57555, GSE66255, GSE76279, and GSE89749. The microarray data presented in this study are available in Appendix A.

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
