# Peer review of "Targeting FGFRs Using PD173074 as a Novel Therapeutic Strategy in Cholangiocarcinoma"

_cancers, 2023, doi:10.3390/cancers15092528_

Round 1

Reviewer 1 Report

This manuscript described the potential usage of PD173074 in treating cholangiocarcinoma (CCA). Bioinformatic analysis suggested the high expression and phosphorylation of FGFRs in CCA patients correlated with poor clinical outcomes (OS and DFS). Authors demonstrated that PD173074 reduces cell viability and induces cell apoptosis in CCA cells, which might be through inhibiting the phosphorylation of FGFRs. Combined treatments of PD173074 and EGF inhibitor (Elotinib) could suppress the proliferation of CCA and stromal cells. In general, this is an interesting study, while some parts need to be improved and detailed explanations should be described. Overall, this manuscript had many typos and was not well presented/described.

Major parts:

1. According to previous in vitro study, the potency of PD173074 against FGFRs is not the highest. How authors obtained enrichment scores of different FGFR inhibitors using Enrichr? This should be mentioned. To confirm the rationale of inhibitor selection, authors could compare PD173074 with other inhibitors, for instance, AZD4547, in this study.

2. Apoptosis assay by visualizing activated caspase3 or cleaved PARP by western blot is suggested.

3. For Fig.5a, FGFR expression by western blot should be included.

4. No further explanation of the results of SynergyFinder. How synergy scores suggested and were compared in Loewe’s additivity, Highest Single Agent (HSA) and Bliss independence mathematical models?

Minor parts:

1. the labelings or characters in all the figures are too small to examine.

2. Page5, line224, "3. D" Tumour Growth Assay; is this a typo?

3. page9, line410, what is Figure 61a?

4. page12, Fig.3c to 3f; "log2FC"; whether "2" should be subscripted?

5. page14, Figure legend in Fig.6 is not complete? Fig.6e - Fig.6j?

Author Response

Reviewer 1:

This manuscript described the potential usage of PD173074 in treating cholangiocarcinoma (CCA). Bioinformatic analysis suggested the high expression and phosphorylation of FGFRs in CCA patients correlated with poor clinical outcomes (OS and DFS). Authors demonstrated that PD173074 reduces cell viability and induces cell apoptosis in CCA cells, which might be through inhibiting the phosphorylation of FGFRs. Combined treatments of PD173074 and EGF inhibitor (Elotinib) could suppress the proliferation of CCA and stromal cells. In general, this is an interesting study, while some parts need to be improved and detailed explanations should be described. Overall, this manuscript had many typos and was not well presented/described.

Major parts:

  1. According to previous in vitro study, the potency of PD173074 against FGFRs is not the highest. How authors obtained enrichment scores of different FGFR inhibitors using Enrichr? This should be mentioned. To confirm the rationale of inhibitor selection, authors could compare PD173074 with other inhibitors, for instance, AZD4547, in this study.

Response: Thank you for bringing this to our attention. We used the web tool Enrichr (https://maayanlab.cloud/Enrichr/enrich) with the search terms "FGFR1," "FGFR2," "FGFR3," and "FGFR4," and utilized the publicly available kinase inhibitor screening database, HMS LINCS KinomeScan (https://lincs.hms.harvard.edu/db/), which contains experimental data from biochemical kinase profiling assays measuring drug binding using a panel of approximately 440 purified kinases in cell-free systems. We understand that this was not clear in the previous version, and we have made the necessary amendments to the manuscript in lines 186-191: "Enrichment of small molecule inhibitors that target FGFR1-4 was performed using Enrichr with the search terms "FGFR1," "FGFR2," "FGFR3," and "FGFR4." The publicly available kinase inhibitor screening database, HMS LINCS KinomeScan (https://lincs.hms.harvard.edu/db/) was used. Statistically significant kinase inhibitors (p<0.05) were ordered according to their combined score, also known as enrichment score, and the top 10 results are reported."

Additionally, in our previous study, we identified five molecular groups in a CCA cohort and found that the FGFR inhibitor PD173074 targeted some subtypes with high FGFR expression (among other receptor tyrosine kinases) [PMID:34577598]. Hence, in this study, we aimed to validate the efficacy of PD173074 in vitro. While we agree that testing and comparing the efficacy of various FGFR inhibitors in CCA would be interesting, it is outside the scope of this study. Furthermore, AZD4547 and BGJ398 have already been extensively studied, and these inhibitors have also been shown to lead to acquired resistance in CCA patients, especially those with FGFR2 fusions. Therefore, we aimed to investigate a novel therapeutic strategy for patients without FGFR2 fusions who are likely to benefit from FGFR inhibition. As such, we limited our investigation to PD173074 to demonstrate its anti-tumor effects in the absence of FGFR2 fusions.

  1. Apoptosis assay by visualizing activated caspase3 or cleaved PARP by western blot is suggested.

Response: Thank you for the valuable suggestion to visualize activated caspase3 or cleaved PARP by western blot for apoptosis assay. We agree that it could be a valuable addition to our study. However, in this study, we have used two quantitative methods to measure apoptosis by PD173074 in vitro, including Annexin V staining via flow cytometry to detect the translocation of phosphatidylserine (PI) to the outer leaflet of the plasma membrane, which is a hallmark of apoptosis. Additionally, we employed a caspase-3 colorimetric assay to quantitatively measure the activity of caspases that recognize the sequence that cleaves PARP (DEVD). The assay is based on the detection of the chromophore p-nitroaniline (p-NA) after cleavage from the labelled substrate DEVD-p-NA. Thus, the results from the caspase 3 assay, in conjunction with the flow cytometry apoptosis assay, demonstrate that PD173034 exerts its cytotoxic effects via apoptosis. While we acknowledge that western blotting could provide further insights into the mechanism of apoptosis, the two quantitative assays used in our study provide quantitative evidence for the apoptotic effect of PD173074 in CCA.

  1. For Fig.5a, FGFR expression by western blot should be included.

Response: Thank you for your valuable suggestion. We agree that it would be interesting to explore the mechanism of individual FGFR receptors and their signaling in CCA. We plan to fully investigate this premise in a follow-up study, along with EGFR receptors, as we found the combinatory inhibition  of these receptors to be efficacious in our current study. However, we decided to focus solely on p-FGFR, which is the activated form of all four functional FGFR receptors (FGFR1-FGFR4). Although there are four individual receptors, the intracellular tyrosine kinase domain is similar in all FGFRs. Amongst the catalytic domain, the tyrosine residues Tyr653 and Tyr654 are the most important for activation of all FGFRs and the subsequent signaling cascades. Currently, there are no other biomarkers besides FGFR2 fusion for FGFR inhibitors. p-FGFR thus poses as a potentially novel biomarker indicative of treatment with FGFR inhibitors.

  1. No further explanation of the results of SynergyFinder. How synergy scores suggested and were compared in Loewe’s additivity, Highest Single Agent (HSA) and Bliss independence mathematical models?

Response: Thank you for bringing this to our attention and we apologise for the oversight. We have amended the results section and added these lines to elaborate the results on synergy score in Lines 422-433: “Synergy was evaluated using three different mathematical models: Loewe’s additivity, Highest Single Agent (HSA) and Bliss independence. The highest synergy score according to Loewe's additivity model was 26.35 in KKU-213 cells at 0.625mM of erlotinib and 2.5mM of PD173074, and 13.62 in RBE cells at 0.625mM of both erlotinib and PD173074. HSA model yielded the highest synergy score of 40.0 in KKU-213 cells at 5mM of both PD173074 and erlotinib, and 19.81 in RBE cells at 5mM of PD173074 and 2.5mM of erlotinib. Bliss independence model showed the highest synergy score of 23.78 in KKU-213 cells at 5mM of both PD173074 and erlotinib, and 10.32 in RBE cells at 5mM of PD173074 and 0.625mM of erlotinib. These results show that these drugs exert their efficacy in a synergistic manner. In addition, our results suggest that dual inhibition with PD173074 and erlotinib may offer a potential therapeutic strategy for treating CCA with a lower concentration of drugs required for maximal effect. These findings highlight the potential of combining PD173074 and erlotinib as a promising therapeutic strategy for CCA treatment.”

Minor parts:

  1. the labelings or characters in all the figures are too small to examine.

Response: Thank you for this comment, we have increased the labels on the figures.

  1. Page5, line224, "3. D" Tumour Growth Assay; is this a typo?

Response: Thank you for bringing this to our attention. We have amended this to “3-D Tumour Growth Assay” in line 231.

  1. page9, line410, what is Figure 61a?

Response: Thank you for bringing this to our attention. We have amended this to “6a,b” in line 419.

  1. page12, Fig.3c to 3f; "log2FC"; whether "2" should be subscripted?

Response: Thank you for bringing this to our attention. We have amended this.

  1. page14, Figure legend in Fig.6 is not complete? Fig.6e - Fig.6j?

Response: Thank you for bringing this to our attention. We have amended this.

Reviewer 2 Report

The author said that FGFR inhibition is a promising therapeutic strategy for Cholangiocarcinoma(CCA)  and p-FGFR can be a useful biomarker for treatment selection and stratification.  This paper covered many experiments and reaction of signalling and explain the mechanism.  This paper was excellent and will be useful of research in therapy of CCA.

Author Response

Reviewer 2:

The author said that FGFR inhibition is a promising therapeutic strategy for Cholangiocarcinoma (CCA) and p-FGFR can be a useful biomarker for treatment selection and stratification.  This paper covered many experiments and reaction of signalling and explain the mechanism.  This paper was excellent and will be useful of research in therapy of CCA.

Response: We thank you for valuable time for reviewing this manuscript and providing comments.

Reviewer 3 Report

Balasubramanian et al., reported that PD173074 a FGFR inhibitor as a potential targeted therapy for cholangiocarcinoma using in-silico and invitro approaches,

1) In fig.2, author said that patients with higher expression of FGFR in CCA displayed better OS using existing datasets, however real-time functional data was missing, the evidence with p-FGFR immunostaining from patients samples were not convincing,

2) In fig.3, KKU-213 and RBE cells displayed better response to cell growth, colony formation, what is the correlation towards protein and gene expression of FGFR1 and FGFR4. Why the inhibitor not showed better response to other cell models, 

3) why author didn't investigated downstream signaling of PI3K activity,

4) It would be more strong supporting evidence, if author would have tried to do some rescue experiments like over expression and knock-down of FGFR in cell models

5) The study has a setback with lack of in-vivo experiments 

Thank you,

Author Response

Reviewer 3:

Balasubramanian et al., reported that PD173074 a FGFR inhibitor as a potential targeted therapy for cholangiocarcinoma using in-silico and invitro approaches,

  • In fig.2, author said that patients with higher expression of FGFR in CCA displayed better OS using existing datasets, however real-time functional data was missing, the evidence with p-FGFR immunostaining from patients samples were not convincing,

Response: We thank you for this comment. As the real-time functional data for the datasets analysed were not readily available, we resorted to use the publicly available TCGA dataset. We agree this is a limitation of small sample size in this study.

  • In fig.3, KKU-213 and RBE cells displayed better response to cell growth, colony formation, what is the correlation towards protein and gene expression of FGFR1 and FGFR4. Why the inhibitor not showed better response to other cell models, 

Response: We selected KKU-100, KKU-213, RBE and TFK-1 cell lines as representative cell lines for this study.  KKU-100 and TFK-1 are established from extrahepatic CCA tumours, whereas KKU-213 and RBE were established from intrahepatic CCA tumours. KKU-100 has a comparatively low expression of FGFRs and has a FGFR3 mutation. TFK-1, KKU-213, and RBE cells were sensitive to PD173074 treatment with IC50 of ~6.6µM, ~8.4µM, and ~11µM, respectively. KKU-100, with low mRNA expression of FGFR was slightly more resistant to PD173074, with an IC50 of ~16µM. As FGFR mutations, apart from FGFR2 fusions, are rare in CCA, KKU-213 and RBE cell lines were selected for further analysis.

3) why author didn't investigate downstream signaling of PI3K activity,

Response: Thank you for this suggestion. We also investigated, p-Akt, p-ERK and p-PLC1G however the western blotting analysis did not yield conclusive results. Only STAT3 was reproducibly decreased. We agree that further in-depth analysis of its downstream activity is important, and we will include it in a follow-up study.

  • It would be more strong supporting evidence, if author would have tried to do some rescue experiments like over expression and knock-down of FGFR in cell models

Response: We thank you for this comment, we have acknowledged this limitation in the discussion section. We agree that it would be interesting to explore the mechanism of individual FGFR receptors and their signalling in CCA. We plan to fully explore this premise in a follow-up study along with other kinase receptors as we found combinations treatment efficacious in this study. For this study, we have stimulated the FGFR receptor using a common FGF-1 ligand and found that it has increased the common p-FGFR site and showed that this phosphorylation can be inhibited by PD173074, highlighting the functional FGFR signalling in our representative cell lines.

5) The study has a setback with lack of in-vivo experiments 

Response: Thank you for this comment, we have validated our findings in in vitro primary lines derived from the patients and these models have proven to be more near-to-patient than animal models. Nonetheless, we agree with the reviewer that this study should be validated in vivo for further translation to clinical setting, however, it is beyond the scope of this funded project. We have added this in the limitations in discussion in Lines 612 – 619 as “While the findings of this study provide valuable insights into the potential therapeutic benefits of the investigated compound, it is important to note that the study has some limitations. One such limitation is that animal work was not included in the research design. We have, however, used near-to-patient primary lines in co-culture with stromal cells for validation. Nonetheless, it is important for future studies to continue to validate the findings using in vivo models, to better understand the potential clinical implications of the compound”.

Round 2

Reviewer 1 Report

1. Decrease in phospho-FGFR should compare with FGFR expression to examine the fractions of FGFR phosphorylation. This rationale is the same as examining the fractions of phospho-STAT3 and ensure the consistency of STAT3 protein level (Fig.5g and 5h).

2. Many typos are still seen in this version. For instance, in the figure legend, where is Figure 2m?

3. The setup of the quadrant in flow-cytometry plots for AnnexinV-PI staining was not convincing so the percentage of apoptotic cells was not reliable. In addition to caspase activity assay, western blot analysis to demonstrate the increase in cell apoptosis is still suggested.

4. Most of the labels in the figures are too small. The authors did not improve and examine all of them.

Author Response

Thank you for taking the time to provide us with your thoughtful comments and suggestions regarding our manuscript. We truly appreciate the thorough evaluation of our work and the insightful feedback you have provided.

We have carefully addressed the issues you have pointed out and made the necessary corrections, including amending the typos in figure legends, increasing the labeling in figures, and providing the full-sized images to the editors to ensure high-quality image resolution.

After careful consideration, we have decided not to perform the additional experiments as suggested, as they are beyond the scope and feasibility of this manuscript. However, we agree with you that these experiments would be valuable and informative, and we have decided to explore the detailed molecular mechanism of these inhibitors in a follow-up study.

Nevertheless, we believe that exploring these aspects in future studies will provide an excellent opportunity to expand upon our current research and improve our understanding of the topic.

Once again, we would like to express our gratitude for your time, effort, and valuable feedback. Your insightful comments have been instrumental in improving the quality and impact of our research.